**Data Availability Statement:** The study data has been submitted for public availability at Dryad, a

# A feasibility study demonstrating that independence, quality of life, and adaptive behavioral skills can improve in children with Down syndrome after using assistive technology

Kaylin White[1]☯, Samuel S. Han 🆔[1]☯*, Angela Britton[2]‡, James Hendrix[2]‡

**1** Research and Development, MapHabit, Inc., Atlanta, Georgia, United States of America, **2** Scientific Research, LuMind IDSC Foundation, Burlington, Massachusetts, United States of America

☯ These authors contributed equally to this work.
‡ AB and JH also contributed equally to this work.
* shan@maphabit.com

## Abstract

Enhancing independence and quality of life are key modifiable outcomes that are short- and long-term goals for children with Down syndrome and for their parents. Here we report the outcome of a 4-week feasibility study in a cohort of 26 children with Down Syndrome, 7–17 years old, who used an assistive technology approach that incorporated smart device software and step-by-step pictures (the MapHabit System). Parents reported improvements in children's activities of daily living, quality of life, and independence. They recommended this technology to other families. This report and its findings underscore the feasibility of using assistive technology in children with Down syndrome within home and family settings. A limiting factor is whether participants who did not complete the study, and thus were not included in analyses, might have impacted the study outcomes. The current findings that assistive technology can be used successfully and effectively in family and home settings set the stage for more informative systematic studies using assistive technology for this population.

**Trial registration:** The clinical trial is registered with ClinicalTrials.gov
Registration number: NCT05343468

## Introduction

Down syndrome (DS) is a leading cause of intellectual disability. Prominent in the phenotype of individuals born with DS is developmentally impaired learning and memory emerging during the early part of their lifespan. Recently, there have been reports of considerable positive impact on quality of life in individuals with Alzheimer's disease (AD) who have used assistive technology (AT) [1]. AT is usually defined as "any piece of equipment, or product system, whether

data repository: (DOI): doi:10.5061/dryad.12jm63z2w.

**Funding:** This manuscript and supporting research were made possible in part by the National Institute on Aging of the National Institutes of Health under award number R43AG065081. This includes 33% ($15,000) funded with federal money and 66% ($30,000) non-government sources. Non-government sources include LuMind IDSC Foundation ($ 15,000) and in-kind donation by MapHabit, Inc. ($ 15,000). The content is solely the responsibility of the authors and does not necessarily represent the official views of the National Institutes of health. The funder MapHabit, Inc. had a role in the decision to publish the research. The funder LuMind IDSC foundation had no role in the conception and execution of the study. LuMind IDSC Foundation https://www.lumindidsc.org/s/1914/20/home.aspx?gid=2&pgid=61 MapHabit, Inc.: https://www.maphabit.com/

**Competing interests:** We have read the journal's policy and the authors of this manuscript have the following competing interests: Co-author, Kaylin White, MS, is a part-time consultant at MapHabit, Inc., the company that developed the assistive technology (the MapHabit System) that is used in this feasibility study. Ms. White receives compensation from MapHabit, Inc. Co-author, Samuel S. Han, B.A., is a part-time consultant at MapHabit, Inc., the company that developed the assistive technology (the MapHabit System) that is used in this feasibility study. Mr. Han receives compensation from MapHabit, Inc. The recruitment of subjects participating in the study was done independently of the authors, and carried out initially by the LuMind IDSC Foundation, a national support entity for families with Down syndrome. Statistical analyses of all data were carried out independently of the authors by a biostatistical resource department of an academic health center. Other than sharing the costs necessary to carry out the study, neither MapHabit, Inc., nor any MapHabit Inc. employees have any financial link to the LuMind IDSC Foundation, or to the independent biostatistical resource used in the study. This does not alter our adherence to PLOS ONE policies on sharing data and materials.

acquired commercially, modified, or customized, that is used to increase, maintain, or improve functional capabilities of individuals with disabilities" [2]. The use of AT within AD communities has been associated with promoting independent living, engagement, enhanced ability to complete activities of daily living, and reduced stress for patients and caregivers [3–6].

While the etiology of cognitive decline due to AD is different than the cognitive deficits due to DS, there are similarities in phenotype characteristics [7]. The findings that the use of AT can assist and support adults with cognitive decline due to AD raise the straightforward question of whether children who have cognitive and behavioral impairments due to DS might also benefit from the use of AT, along with their caregivers. Importantly, the efficacy of AT in children with DS is an area of research that is still in its infancy and expanding with the rise of mobile device technology and its subsequent accessibility and familiarity [8]. In turn, continued empirical investigations of using AT among the DS community are needed. To address this need, the current study examined the feasibility of an AT as an intervention among children with DS. For the purpose of this study, we use the term feasibility to mean an assessment of a future randomized controlled trial evaluating the effect of an intervention" [9]. Here, we describe the findings from a feasibility study that used a simple pre-post intervention design to assess whether there was behavioral impact in a group of children with DS, ages 7–17, after using an AT for four weeks.

## Methods

### Clinical trial registration, consent, and enrollment

**Ethical approval and clinical trial registration.** Advarra Institutional Review Board (IRB) served as the IRB of record (Pro00039611). The study is registered as a clinical trial with ClinicalTrials.gov (NCT05343468), a registry that meets the requirements of the International Committee of Medical Journal Editors (ICMJE). Registration for the study did not occur prior to participant enrollment due to the intervention being at the feasibility stage and the device not meeting the criteria for regulatory oversight by the Food and Drug Administration. The study underwent appropriate registration per the request that all forms of interventional studies be registered through an ICMJE-approved registry. The authors confirm that all ongoing and related trials for this intervention are registered.

**Recruitment, consent, and enrollment.** Community outreach activities (social media, email, and web site posting) were performed by a Down syndrome research and resources organization based in the United States. Interested parties were directed from the organization channels to clinical coordinators of the study. Identified candidates completed the informed consent process, which included an electronic informed consent form that was filled out by the parent or legal guardian of the child with Down syndrome through a virtual meeting with a clinical coordinator. Informed consent forms were electronically signed, documented, and stored in REDCap, a research electronic data capture database. Assent forms were provided to the parents and legal guardians to explain the study and obtain consent from the children. The clinical coordinators worked with the families for the duration of the virtual study.

Parents or guardians are referred to here as "caregivers" and the enrolled children as "participants". Caregivers completed all the assessments from which the findings were derived. Participants used the AT under the supervision of their caregiver throughout the duration of the study. Monetary compensation was provided to families for completing the requirements of the study. Eligibility criteria included the children being diagnosed with Down syndrome and being between the ages of 7 to 17. Additionally, participants were required to have internet access and be proficient in English. 72 families were assessed for eligibility. 18 of those families were excluded, with two of them not meeting inclusion criteria and the rest ultimately

declining to participate. 54 participants began the intervention, with 23 of them lost after no contact from them during their participation and five participants who expressed no desire to continue. A total of 26 families fully completed the study (Table 1 and Fig 1). The study began recruitment and enrollment in September of 2020 and reached a primary completion date of June of 2021. The final study completion date was in August of 2021.

## The MapHabit system

Most AT that are dedicated to individuals with DS assist in teaching and reinforcing highly specific functional skills, such as memory, social inclusion, or labor [10–12]. Although these technologies each focus on particular skills, what is universal is the aim to support decision-making capacity, increase self-determination, and enhance self-esteem through daily activities in individuals with DS, all of which are factors that should be prioritized in AT development [8]. The MapHabit System (MHS) is a commercially available visual mapping software application that utilize visual, audio, and text media to create step-by-step visual guides—which are called maps—to assist individuals and their caregivers in structuring and accomplishing activities of daily living (ADLs) (Fig 2). The MHS has been shown to be effective in enhancing overall quality of life in individuals with AD [3–6]. The goal of the application is to develop and facilitate habits and routines using structured visual and auditory stimuli that can be customized by the user and can include educational and lesson-based material in addition to ADLs. This allows the user to engage in a wide range of functional skills, a unique aspect that differs from other AT that emphasize a specific set of functional skills. Moreover, the software application attempts to enhance the independence and confidence of their users [3–6], aligning with the tenants of AT mentioned above. The application was made available to families through compatible smartphones and tablets.

## Study design

Participants and caregivers were first trained on the use of the MHS by study coordinators in a single session, including discussing the most needed ADL maps for the participants and how to make and personalize their own maps. These training sessions were approximately one hour long. Caregivers were instructed to create and implement three visual maps for the

**Table 1. Participant demographics.**

| Variables | Sample ($n$ = 26) |
|---|---|
| Age | |
| Mean ± SD | 11.5 ± 3.4 |
| Min-Max | 7.0–17.0 |
| Gender (%) | |
| Female | 11 (42.3) |
| Male | 15 (57.7) |
| Race (%) | |
| White or Caucasian | 20 (76.9) |
| Hispanic or Latinx | 3 (11.5) |
| Black or African American | 1 (3.8) |
| Other | 2 (7.7) |

Description of the demographics of all participants included in the study. Spearman's rank order correlation test was used for any correlations between demographic variables and assessment scores. No significant correlations were observed between any variables (all $p$ values >1.0).

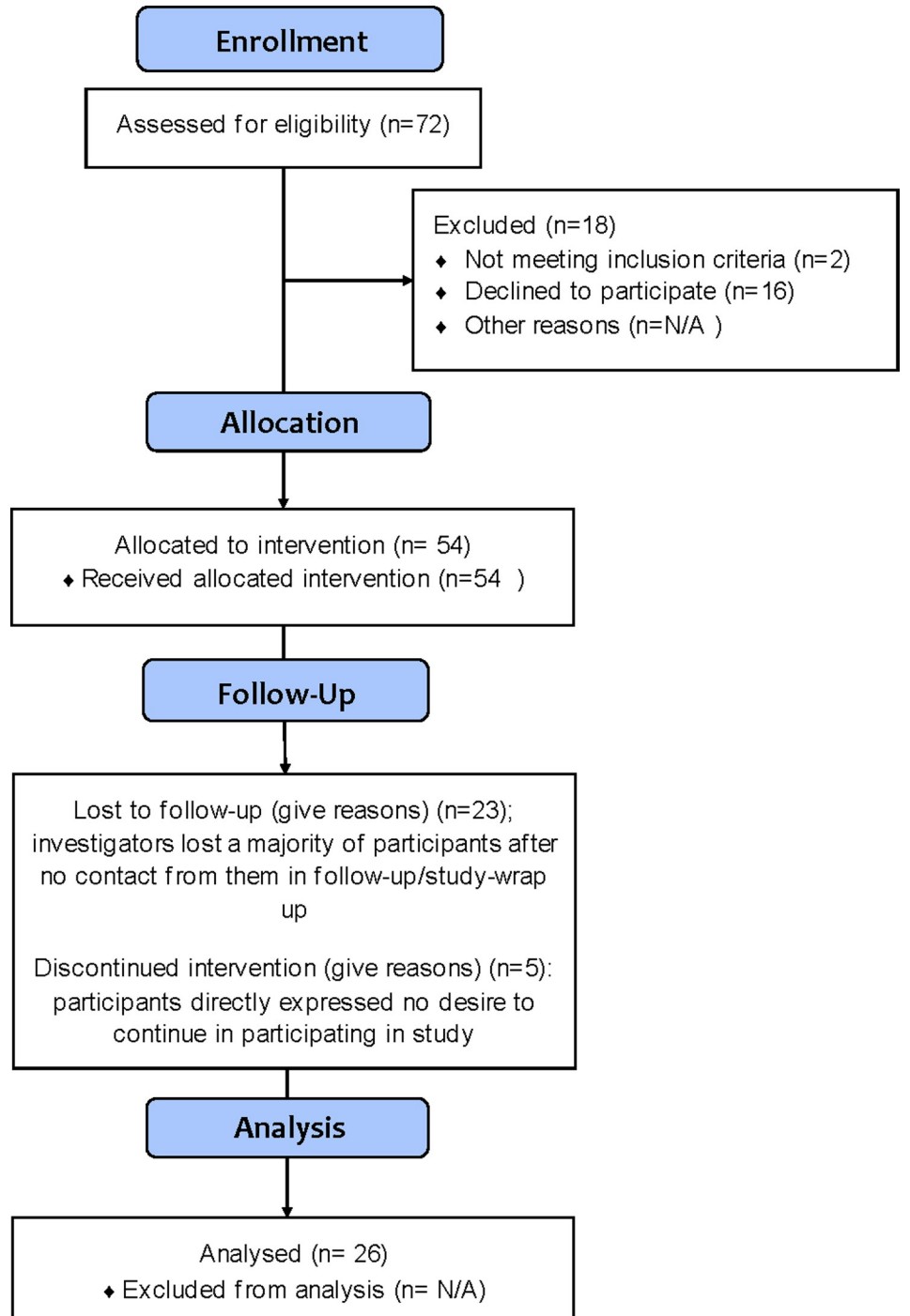

**Fig 1. CONSORT participant flow diagram.**

participants, and to use the application at least 5 days per week. Caregivers were administered standardized assessment questionnaires before and after four weeks of using the MHS. This single-arm repeated measures design permitted us to obtain baseline measures and compare them with the post-intervention outcomes. All training and pre/post assessments between caregivers and the study coordinators were done remotely via virtual video calls.

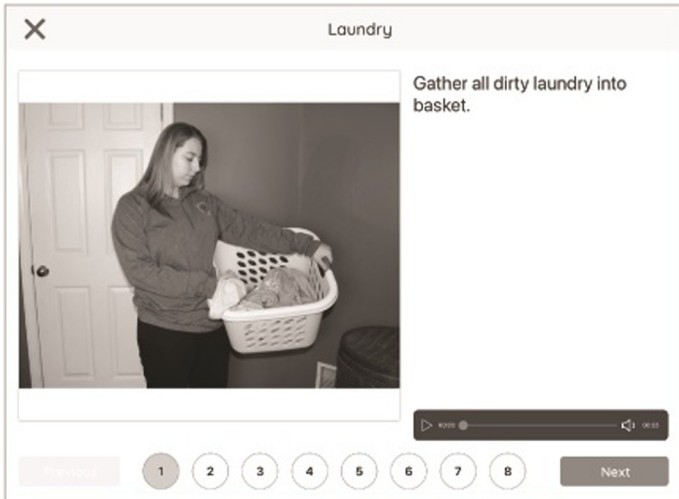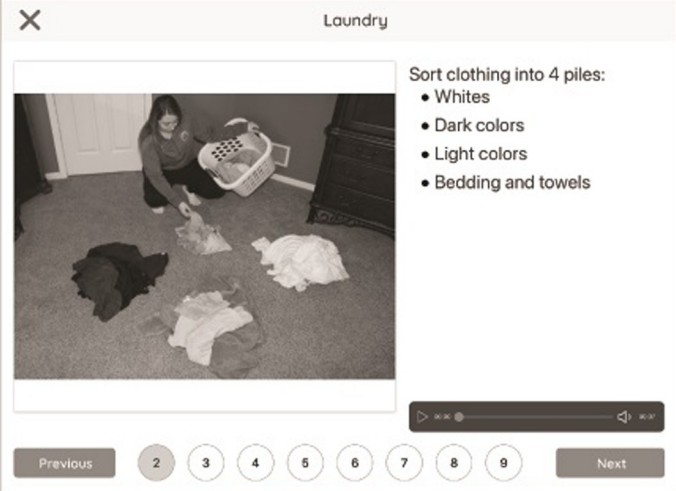

**Fig 2. Example of a visual map.** An example of a visual map for an activity of daily living titled "Laundry", in which there are a total of 8 sequential steps guiding the user through a certain aspect of the activity. Each step appears individually and only progresses to the next step once the user interacts with the device's touch screen. Once the user reaches the last step of the activity, the user is able to confirm that it has been successfully completed. The individual pictured in Fig 2 has provided written informed consent (as outlined in PLOS consent form) to publish their image alongside the manuscript. Republished under a CC BY license, with permission from MapHabit, Inc., original copyright 2022.

## Assessments

Caregivers completed the pre/post assessments of their participants using the Adaptive Behavior Assessment System Third Edition (ABAS-3) [13] to evaluate conceptual, social, and practical behavioral skills that are significant for day-to-day functioning [14]. Analyses were carried out using the ABAS-3 General Adaptive Composite (GAC), a final score that captures all three of an individual's adaptive behavioral skills. At the end of the study, the ABAS-3 was re-administered to caregivers, along with two additional assessments. First, an 18-item quality-of-life questionnaire (QoL-18) that evaluates a range of participants' behaviors, including mood, engagement, and memory at the end of the study compared to before the use of the MHS [3–5], which used a 5-point Likert scale format [15] where a higher score is better (5. Much better, 4. Better, 3. Not much change, 2. Worse, 1. Much worse). Secondly, a 2-item Satisfaction Scale (SS-2) that quantified caregivers' endorsements to two survey questions: How satisfied were they with the MHS? Would they recommend MHS to others?

## Data analyses

All analyses were carried out independently by an academic university's Department of Biostatistics, not connected with MapHabit, Inc. or any study authors. Statistical Product and Service Solutions (SPSS) statistics software, developed by International Business Machines Corporation (IBM), was used for all analyses. Paired group comparisons used the nonparametric Wilcoxon matched-pairs signed ranks test, and unpaired comparisons used the nonparametric Mann-Whitney test. All comparisons used 2-tail tests. For Quality-of-Life item analyses, the one-sample Wilcoxon test (also referred to as one-sample Wilcoxon signed-rank test) was used to compare scores against the null hypothesis (score of '3').

# Results

## Demographics

Spearman's rank order correlation test was used to analyze for any associations between the demographics of participants and their respective scores on the assessments. No correlations of significance were found between any of the participants' demographic variables (Table 1) and scores on any the pre and post assessment variables (all $p$ values $> 1.0$).

## ABAS-3

GAC1 refers to pre-scores. GAC2 refers to post-scores. The 26 participants obtained a median GAC2 score of 48.0 at the end of training, 5.5 points (8.5%) higher than their median GAC1 score of 42.5 ($p = 0.0071$; Table 2). Eighteen participants (69.2%) showed improved scores (GAC1 = 39.5; GAC2 = 53.0; ($p = 0.001$), while eight participants (30/7%) showed lower scores (GAC1 score = 47.5; GAC2 score = 40.5; $p = 0.0078$). There was no significant difference between the eighteen and eight participants in median GAC1 scores (18 participants = 39.5, 8 participants = 47.5, ($p = 0.90$)). For the pre-post comparisons, Wilcoxon matched pairs signed rank test was used. The Mann-Whitney U test was used to compare between groups.

## QOL-18

Scores for 13 of the 18 items reached statistical significance (i.e., they were significantly higher than the null hypothesis score of 3.0; $p$ values of the one-sample Wilcoxon test ranged from $< 0.05$ to $< 0.001$; Fig 3). Five items had scores not significantly different from 3.0(all $p$ values $> 0.05$).

## SS-2

The response scores of the 26 caregivers on each of the two questions were overwhelmingly positive with 20 caregivers (76.9%) "strongly agree" or "agree" in answers to both questions. Only two caregivers had negative responses to the two questions.

# Discussion

## ABAS-3

While the overall pre-post improvement between GAC1 and GAC2 scores was numerically small (Table 2), the level of statistical significance was high ($p = 0.0071$), even though not all participants showed improved GAC2 scores (see Results). Scores on the QOL-18 and the SS-2, likewise, reflected overall strong positive experiences. Taken together, these findings suggest

**Table 2. ABAS-3 pre and post scores.**

| Group | Median (IQR) | | $p^I$ |
|---|---|---|---|
| | GAC1 | GAC2 | |
| All ($n = 26$) | 42.5 (25) | 48.0 (27) | <0.01 |
| Improved ($n = 18$) | 39.5 (31) | 53.0 (32) | <0.001 |
| Decreased ($n = 8$) | 47.5 (14) | 40.5 (16) | <0.01 |

Note: IQR indicates interquartile range

[1] $p$ values were calculated using the Wilcoxon matched pairs signed rank test; $p<0.05$ statistically significant; $p<0.001$ highly significant

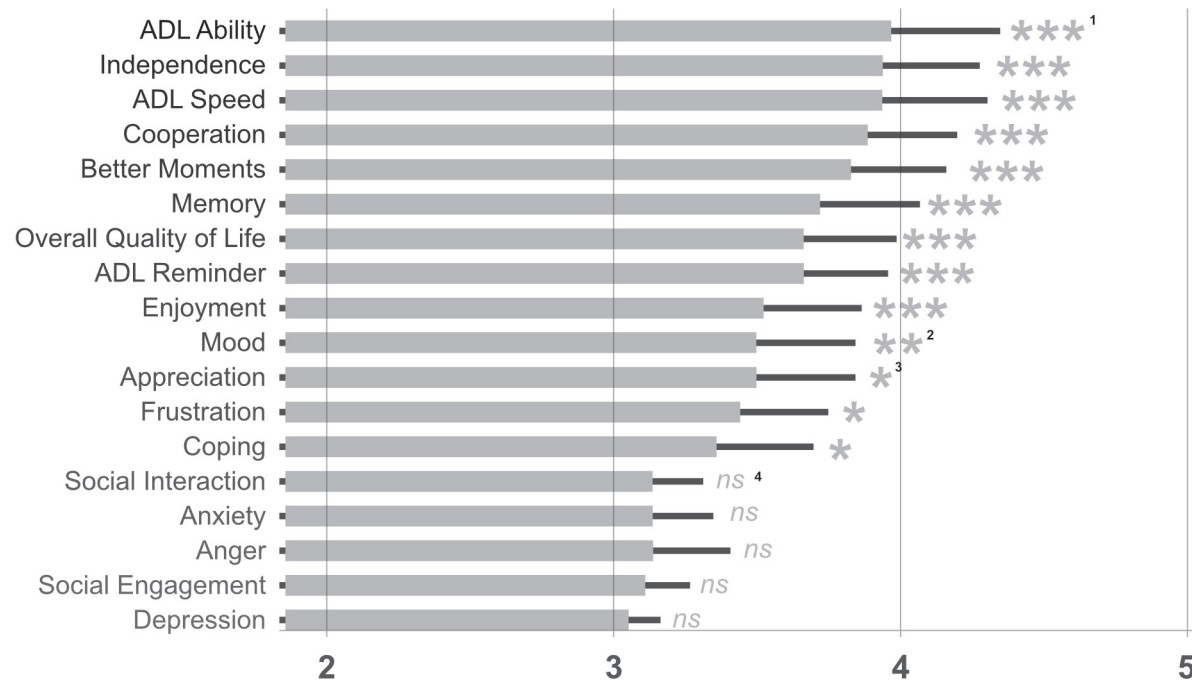

Note: One-sample Wilcoxon test was used for analysis

━ Black lines indicate standard deviation

1 *** indicates $p < .001$

2 ** indicates $p < .01$

3 * indicates $p < .05$

4 *ns* indicates no significance

**Fig 3. Quality of life scores.** Average scores for the 26 participants on the 18-item Quality of Life questionnaire. Words on the left abbreviate the full question (e.g., ADL Ability: "Compared to before using the MapHabit System, is your child's ability to complete activities of daily living now...") and caregivers answered using the 5-point Likert scale that ranged from 1 = much worse to 5 = much better (see Methods).

that the pre-post difference on the ABAS-3 assessment, although numerically small, represented a meaningful and positive behavioral change.

### QOL-18

Thirteen of the 18 items were endorsed by caregivers as having significant positive changes (all $p's < 0.01$), including items related to accomplishing ADLs as well as items focused on other behaviors and on emotions. Only five questions did not evidence change, ($p's > 0.05$; Fig 3). Caregivers frequently and spontaneously commented on how the use of the MHS improved their own quality of life (e.g., reduced stress, caregiver burden) as well as that of their participants.

### SS-2

One of the challenges to the adoption of AT in various communities is simply ignorance of its availability [16]. The majorly positive responses to the experiences using the MHS by

caregivers in the present study are encouraging. Enhancing communication and understanding of availability seems a small hurdle toward more universal use of AT.

## Conclusion

The findings observed in this study underscore the feasibility of using AT and it having a positive impact on behavioral outcomes in children with DS within family settings. Moreover, the findings set the stage for conducting systematic RCT studies to further evaluate the efficacy of AT. To underscore the latter point, the present feasibility study did identify methodological limitations that can inform future research of the MHS. First, a RCT would allow for a more systematic assessment of variables that could impact behavioral change. Second, eight participants showed no improvement on ABAS-3 scores, and it would be useful to explore what variables (e.g., parent engagement, grouping of participants based on various other behavioral phenotypes) might be linked to positive and negative outcomes. Third, our limited sample size prevented meaningful exploration of the effects of demographic variables such as age, gender, ethnicity, or additional intellectual disabilities, as well as more systematic subgroup analyses. A good example is the ABAS-3 scores of the 18 participants who reported improvement versus the 8 participants who did not. Retrospective power analysis revealed that in order to achieve a statistical power of 0.80 to detect a 95% confidence ($p = 0.05$), a total sample size of 30 would be required. Our recruited sample size ($n = 26$) for this current feasibility study was underpowered for stratification into subgroups and points to the important consideration that researchers must be prepared to devote considerable time and effort to obtain adequate levels of recruitment of Down syndrome families. Specifically, it will be important for future investigations to stratify participants with DS based on the severity of their intellectual disability. This specificity can allow a deeper understanding of this AT's effectiveness in the outcomes that were observed here and address the wide range of intellectual or developmental disability that is observable in the DS community. In turn, such investigation can expand to observe if AT has a similar impact on other intellectual and developmental disability communities, such as families with autism. Fourthly, we acknowledge that 28 participants did not complete the study and resulted in incomplete data. This may have been due to the study having been taken place during the beginning of the COVID-19 pandemic and subsequent nationwide lockdowns in the United States. This was a time of uncertainty met with many challenges as families were navigating quarantine protocols, school shutdowns, and lifestyle changes, leading to the inability to continue with their participation in the study. Whether the incomplete participation of these subjects could have affected the study's outcomes is another limiting factor. Lastly, the study was not designed to focus on caregiver stress or burden variables, and the impact of these variables will surely be useful to explore, as will having participants complete their own assessments.

Previously published work within this field explored the efficacy of assistive technology for children with Down Syndrome under the context of school learning or particular functional skills, but the current findings that assistive technology can be used successfully and effectively in family and home settings set the stage for more informative systematic studies. Overall, these findings are encouraging. The use of AT may provide important and effective interventions to help enhance overall quality of life and independence for children living with DS as well as their parents or caregivers. Moreover, as described in the Introduction, AT has experienced growing acceptance within the AD community as a useful intervention for helping maintain QOL and independence. The availability of a similar kind of effective intervention early in life in DS may prove advantageous for the well-being and continuity of care for the intellectual disability community.

## Supporting information

**S1 Protocol. Protocol of current study.**
(DOCX)

**S1 Checklist. TREND statement checklist.**
(DOC)

## Acknowledgments

This manuscript and supporting research were made possible in part by the support and participation of families within the Down syndrome community. We thank Kevin Xu and Yanan Wang for assistance and validation of statistical analyses, Paolo Aguila for assistance with figure illustrations, and Matthew Golden and Stuart Zola, co-Founders of MapHabit, Inc., and Hampus Hillerstrom, President and CEO of Lumind IDSC Foundation for their support that enabled the undertaking of this study.

## Author Contributions

**Conceptualization:** Samuel S. Han, Angela Britton, James Hendrix.

**Data curation:** Samuel S. Han.

**Investigation:** Kaylin White, Samuel S. Han.

**Methodology:** Kaylin White, Samuel S. Han.

**Project administration:** Kaylin White, Samuel S. Han.

**Supervision:** Kaylin White, Samuel S. Han.

**Writing – original draft:** Kaylin White, Samuel S. Han, Angela Britton, James Hendrix.

**Writing – review & editing:** Samuel S. Han, Angela Britton, James Hendrix.

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
