## [Decision Letter · Decision Letter 0]

1 Sep 2022

PONE-D-22-05428

Independence, Quality of Life, and Adaptive Behavioral Skills Improved in Childrenwith Down Syndrome After Using Assistive Technology

PLOS ONE

Dear Dr. Han,

Thank you for submitting your manuscript to PLOS ONE. After careful consideration, we feel that it has merit but does not fully meet PLOS ONE’s publication criteria as it currently stands. Therefore, we invite you to submit a revised version of the manuscript that addresses the points raised during the review process.

The manuscript has been evaluated by one reviewer, and his comments are available below.

The reviewer has raised a number of major concerns that need attention. He  requests additional information on methodological aspects of the study and revisions to the statistical analyses.

Could you please revise the manuscript to carefully address the concerns raised?

We look forward to receiving your revised manuscript.

Kind regards,

Lorena Verduci

Staff Editor

PLOS ONE

“This manuscript and supporting research were made possible in part by the support and participation of families within the Down syndrome community, and in part by the National Institute of Aging of the National Institutes of Health under award number RAG065081A. This includes 33% ($15,000) funded with federal money and 66% ($30,000) non-government sources. Non-government sources include LuMind IDSC Foundation ($ 15,000) and in-kind donation by MapHabit, Inc. ($ 15,000). The content is solely the responsibility of the authors and does not necessarily represent the official views of the National Institutes of health. We thank Kevin Xu and Yanan Wang for assistance and validation of statistical analyses, Paolo Aguila for assistance with figure illustrations, and Matthew Golden and Stuart Zola, co-Founders of MapHabit, Inc., and Hampus Hillerstrom, President and CEO of Lumind IDSC Foundation for their support that enabled the undertaking of this study.”

“This manuscript and supporting research were made possible in part by the National Institute of Aging of the National Institutes of Health under award number RAG065081A. This includes 33% ($15,000) funded with federal money and 66% ($30,000) non-government sources. Non-government sources include LuMind IDSC Foundation ($ 15,000) and in-kind donation by MapHabit, Inc. ($ 15,000). The content is solely the responsibility of the authors and does not necessarily represent the official views of the National Institutes of health. The funder MapHabit, Inc. had a role in the decision to publish the research. The funder LuMind IDSC foundation had no role in the conception and execution of the study.

LuMind IDSC Foundation: https://www.lumindidsc.org/s/1914/20/home.aspx?gid=2&pgid=61

MapHabit, Inc.: https://www.maphabit.com/”

“I have read the journal's policy and the authors of this manuscript have the following competing interests:

Co-author, Kaylin White, MS, is a part-time consultant at MapHabit, Inc., the company that developed the assistive technology (the MapHabit System) that is used in this feasibility study. Ms. White receives compensation from MapHabit, Inc.

Co-author, Samuel S. Han, B.A., is the Clinical Lead of MapHabit, Inc., the company that developed the assistive technology (the MapHabit System) that is used in this feasibility study. Mr. Han receives salary from MapHabit, Inc.

The recruitment of subjects participating in the study was done independently of the authors, and carried out initially by the LuMind IDSC Foundation, a national support entity for families with Down syndrome. Statistical analyses of all data were carried out independently of the authors by a biostatistical resource department of an academic health center. Other than sharing the costs necessary to carry out the study, neither MapHabit, Inc., nor any MapHabit Inc. employees have any financial link to the LuMind IDSC Foundation, or to the independent biostatistical resource used in the study.”

Reviewers' comments:

Reviewer's Responses to Questions

**Comments to the Author**

1. Is the manuscript technically sound, and do the data support the conclusions?

Reviewer #1: Partly

2. Has the statistical analysis been performed appropriately and rigorously? 

Reviewer #1: No

3. Have the authors made all data underlying the findings in their manuscript fully available?

Reviewer #1: Yes

4. Is the manuscript presented in an intelligible fashion and written in standard English?

Reviewer #1: Yes

5. Review Comments to the Author

Reviewer #1: The study aims to assess the significant behavioral impact of Assisted Technology on a group of children with Down Syndrome ages 7-17.

Comments

If it is a feasibility type of study, the word feasibility is to be added to the title.

Data analyses section is to be placed in the method section and not in the results section.

Line 115, the publisher name for the statistical software and its version to be stated. The level of statistical significance acceptance is to be stated.

Line 116-117, the use of the statistical tests for what comparison in the context of this study and the name of the correlation test is to be clearly stated.

Effect size could be employed to determine the effect of the intervention.

For a feasibility study, significance testing is not encouraged unless the sample size is properly/adequately powered (Line 166, 'limited sample size')

Results

The percentage figures are to be presented with at least one decimal point.

Line 117-119, Line 120-125, the data to be presented in table form.

Figure 2A could be presented in table form with median and IQR values rather than using a bar chart.

Line 122-123, the flow of the sentence requires improvement.

Line 124, ‘two subgroups participants’ to be revised. To state ‘there was no statistical difference between the 18 and 8 participants.'

Line 127, the name of the statistical test to be denoted in the Figure 2b footnote and in the data analyses section.

Figure 2a, statistical test to be denoted in the figure footnote. P value could be displayed on the figure rather than mentioned in the title.

Figure 2b, *, *** and dark line to be denoted in the Figure footnote.

CONSORT diagram to be revised and tweaked based on this study e.g. method, flow etc

Not all references comply with the journal format.

6. PLOS authors have the option to publish the peer review history of their article (what does this mean?). If published, this will include your full peer review and any attached files.

Reviewer #1: No

---

## [Author Response · Author response to Decision Letter 0]

3 Nov 2022

Dear Dr. Verduci, 

Thank you for giving us the opportunity to submit a revised draft of our manuscript previously titled “Independence, Quality of Life, and Adaptive Behavioral Skills Improved in Children with Down Syndrome After Using Assistive Technology” to PLOS ONE, now titled “A Feasibility Study Demonstrating that Independence, Quality of Life, and Adaptive Behavioral Skills Can Improve in Children with Down Syndrome After Using Assistive Technology” (see Reviewer Comment 1). We are grateful for the time and effort that you and the reviewer have dedicated in providing valuable feedback on our manuscript. We appreciate the insightful comments from you and the reviewer. We were able to incorporate changes to reflect the suggestions that were provided. Below are the point-by-point responses to the editor and reviewer’s comments and concerns: 

Comments from Editor

Comment 1: Please ensure that your manuscript meets PLOS ONE's style requirements, including those for file naming

Response: Thank you for this reminder. We have reviewed the manuscript and made changes that meet PLOS ONE’s style requirements.

Comment 2: Please provide additional details regarding participant consent. In the ethics statement in the Methods and online submission information, please ensure that you have specified what type you obtained (for instance, written or verbal, and if verbal, how it was documented and witnessed). If your study included minors, state whether you obtained consent from parents or guardians. If the need for consent was waived by the ethics committee, please include this information.

Response: We have added the details mentioned in the comment above into the Methods section of the revised manuscript. 

Comment 3: We note that you have provided funding information that is not currently declared in your Funding Statement. However, funding information should not appear in the Acknowledgments section or other areas of your manuscript. We will only publish funding information present in the Funding Statement section of the online submission form… Please remove any funding-related text from the manuscript and let us know how you would like to update your Funding Statement… Please include your amended statements within your cover letter; we will change the online submission form on your behalf.

Response: We have removed the funding information from the Acknowledgements section and included our amended statements within our new cover letter. 

Comment 4: Please confirm that this does not alter your adherence to all PLOS ONE policies on sharing data and materials, by including the following statement: ""This does not alter our adherence to PLOS ONE policies on sharing data and materials.” (as detailed online in our guide for authors http://journals.plos.org/plosone/s/competing-interests). If there are restrictions on sharing of data and/or materials, please state these. Please note that we cannot proceed with consideration of your article until this information has been declared. Please include your updated Competing Interests statement in your cover letter; we will change the online submission form on your behalf.

Response: We confirm this does not alter our adherence to PLOS ONE policies on sharing data and materials. The updated Competing interests statement is included in our amended statements within our new cover letter. 

Comment 5: We note that you have stated that you will provide repository information for your data at acceptance. Should your manuscript be accepted for publication, we will hold it until you provide the relevant accession numbers or DOIs necessary to access your data. If you wish to make changes to your Data Availability statement, please describe these changes in your cover letter and we will update your Data Availability statement to reflect the information you provide.

Response: We confirm the data availability statement upon acceptance. 

Comment 6: Please include your full ethics statement in the ‘Methods’ section of your manuscript file. In your statement, please include the full name of the IRB or ethics committee who approved or waived your study, as well as whether or not you obtained informed written or verbal consent. If consent was waived for your study, please include this information in your statement as well.

Response: The full ethics statement has been included in the ‘Methods’ section of our manuscript file. 

Comments from Reviewer

 Comment 1: If it is a feasibility type of study, the word feasibility is to be added to the title.

Response: The title has been revised to address this comment.

Comment 2: Data analyses section is to be placed in the method section and not in the results section.

Response: The suggested change has been made in the manuscript.

Comment 3: Line 115, the publisher’s name for the statistical software and its version to be stated. The level of statistical significance acceptance is to be stated.

Response: The suggested additions have been made in the manuscript. 

Comment 4: Line 116-117, the use of the statistical tests for what comparison in the context of this study and the name of the correlation test is to be clearly stated.

Response: The suggested additions have been made in the manuscript. 

Comment 5: Effect size could be employed to determine the effect of the intervention.

Response: See Response to Comment 6, below.

Comment 6: For a feasibility study, significance testing is not encouraged unless the sample size is properly/adequately powered (Line 166, 'limited sample size')

Response: We have added a section to the discussion indicating that our study was underpowered for some analyses (e,g., further stratification of subject groups, effect size). Nevertheless, as we have indicated, the findings evidenced important behavioral outcomes to be further explored, as well as the need for researchers to be prepared to devote considerable resources to recruitment efforts and participant retention.

Comment 7: The percentage figures are to be presented with at least one decimal point.

Response: The suggested change has been made in the manuscript.

Comment 8: Line 117-119, Line 120-125, the data to be presented in table form.

Response: The suggested change has been made in the manuscript. 

Comment 9: Figure 2A could be presented in table form with median and IQR values rather than using a bar chart.

Response: The suggested change has been made in the manuscript. 

Comment 10: Line 122-123, the flow of the sentence requires improvement.

Response: The suggested clarification has been made in the manuscript. 

Comment 11: Line 124, ‘two subgroups participants’ to be revised. To state ‘there was no statistical difference between the 18 and 8 participants.'

Response: The suggested revision has been made in the manuscript. 

Comment 12: Line 127, the name of the statistical test to be denoted in the Figure 2b footnote and in the data analyses section.

Response: The suggested additions has been made in the manuscript and figure. 

Comment 13: Figure 2a, statistical test to be denoted in the figure footnote. P value could be displayed on the figure rather than mentioned in the title.

Response: The suggested additions and changes have been made in the manuscript and figure. 

Comment 14: Figure 2b, *, *** and dark line to be denoted in the Figure footnote.

Response: The suggested additions have been made in the figure. 

Comment 15: CONSORT diagram to be revised and tweaked based on this study e.g., method, flow etc.

Response: We request that the CONSORT diagram that was previously submitted be removed. In PLOS ONE’s submission guidelines, the following statement is included: “Clinical trials must be reported according to the relevant reporting guidelines, i.e. CONSORT for randomized controlled trials, TREND for non-randomized trials, and other specialized guidelines as appropriate.” As our study is non-randomized, we believe the TREND statement checklist is appropriate. We have included a revised TREND statement in our resubmission. 

Comment 16: Not all references comply with the journal format.

Response: We have adjusted the references to comply with the journal format. 

We look forward to hearing from you in due time regarding our submission and to respond to any further questions and comments you may have.

---

## [Decision Letter · Decision Letter 1]

10 Jan 2023

PONE-D-22-05428R1A feasibility study demonstrating that independence, quality of life, and adaptive behavioral skills can improve in children with Down syndrome after using assistive technologyPLOS ONE

Dear Dr. Han,

Thank you for submitting your manuscript to PLOS ONE. After careful consideration, we feel that it has merit but does not fully meet PLOS ONE’s publication criteria as it currently stands. Therefore, we invite you to submit a revised version of the manuscript that addresses the points raised during the review process.

Please see the reviewers' comments below. When revising your manuscript, please ensure you fully address each of the reviewers' comments; in particular, please provide a detailed response to Reviewer #3's comment regarding the evaluation of the study participants' intellectual disability, and to the comments regarding clarity and presentational structure from Reviewer #2.

We look forward to receiving your revised manuscript.

Kind regards,

Hugh Cowley

Staff Editor

PLOS ONE

Reviewers' comments:

Reviewer's Responses to Questions

**Comments to the Author**

1. If the authors have adequately addressed your comments raised in a previous round of review and you feel that this manuscript is now acceptable for publication, you may indicate that here to bypass the “Comments to the Author” section, enter your conflict of interest statement in the “Confidential to Editor” section, and submit your "Accept" recommendation.

Reviewer #1: All comments have been addressed

Reviewer #2: (No Response)

Reviewer #3: (No Response)

2. Is the manuscript technically sound, and do the data support the conclusions?

Reviewer #1: Partly

Reviewer #2: Partly

Reviewer #3: Yes

3. Has the statistical analysis been performed appropriately and rigorously? 

Reviewer #1: Yes

Reviewer #2: No

Reviewer #3: Yes

4. Have the authors made all data underlying the findings in their manuscript fully available?

Reviewer #1: Yes

Reviewer #2: Yes

Reviewer #3: (No Response)

5. Is the manuscript presented in an intelligible fashion and written in standard English?

Reviewer #1: Yes

Reviewer #2: No

Reviewer #3: Yes

6. Review Comments to the Author

Reviewer #1: (No Response)

Reviewer #2: Interesting and important study!

I appreciate reading the comments from the first reviewer and responses of the authors, which have improved the paper and made it clearer. Although I do have comments of my own.

Abstract: The term ’nationally’ implies a reader knows where the study is conducted. Consider naming the country instead.

Note: Personally, I find it difficult to use the framing “the first study on X”, because 1) How do you know that unless you have systematically searched the literature (as in a systematic review), 2) it puts a time stamp on the study, making it less interesting once another study on the same topic is published. It adds no scientific value, because there are thousands of studies that are first on a topic.

Introduction/Research question: I don’t generally approve revising the wording in a research question (although the question you have is very vague) but I think the term significant should not be stated in the question. Is that the ‘test’ you think would answer whether AT is feasible or not? Another statistical approach would also serve as equally good (eg., Bayesian statistics).

Introduction: I think the outline lacks a clear description of what is meant by feasibility of AT. There are no standards mentioned, conditions to be met (from models, theories, checklists) of what would count as a positive use of AT in everyday life. Additionally, there is little to no description of what kind of AT that is the focus of the study, and why you choose to study the MapHabit System over other technologies, and why the chosen technology may be a good fit for people with Down’s syndrome (other than resemblance to Alzheimer’s). Quality of life is also not mentioned.

With this being said, many of the points I have raised are presented in the Methods section, and I think the outline of the paper is a bit confusing. I would have preferred describing and provide the rationale for each construct/the feasibility take in the Introduction section, which would make the Method section follow more logically. I am not familiar with this standard/format or what it is called.

Method section

General: the Enrollment, consent, & clinical trial registration section would be a lot easier to read if there were subheadings.

L54: Name the country

L115: You write: “This single-arm repeated measures design permitted the participants to serve as their own controls.” No, I don’t think the design allow for them being their own controls just because you measure pre- and post intervention (look up the designs in eg., Shadish, Cook and Campbell). Eg., in single subjects design you have multiple measurements to be able to observe a change more reliably bc a change in any of the measurements may be due to other reasons than the intervention. Consider removing that argument or provide a clearer description, number of measurement points etc.

L130-135: Provide rational of why you used non-parametric tests. They are less sensitive than parametric tests for observing change, for example.

Additionally: You write “1-sample t-test was used to compare scores against the null hypothesis (score of ‘3’). I am confused”. A one-sample t-test can be used to compare a point estimate (from your sample) to a fixed value. Consider explaining this more clearly, eg., if you test against the score 3. Also, in the results section you write: “one-sample Wilcoxon test”. Is this correct?

Method and Results

You have not mentioned the attrition or analysed what that meant for your conclusions. I think this is important because 23 participants were ‘lost’ and five expressed no desire to continue. The sample you have analysed are the ones that are ‘sufficiently satisfied’ because they fulfilled the trial, I guess. On the positive side, you analyse those that did not progress, but I think a general take is needed that includes all participants entering the study.

I think a conflict of interest statement should be added to the manuscript.

Explanation for the review questions with a 'No'

Q2, 3: If authors adress my comments these will be turned into a 'Yes'

Q5: Language is fine, but I have questions on the format/outline of the paper. If adequately met, this will also turn into a 'Yes'

Reviewer #3: This well written research paper focuses on the use of AT from children with DS - It is suggested authors to give more information about the evaluation of their intellectual disability; the results may differ according the severity

7. PLOS authors have the option to publish the peer review history of their article (what does this mean?). If published, this will include your full peer review and any attached files.

Reviewer #1: No

Reviewer #2: No

Reviewer #3: **Yes: **Aspasia Serdari

---

## [Author Response · Author response to Decision Letter 1]

17 Feb 2023

Dear Mr. Cowley, 

Thank you for giving us the opportunity to submit a revised manuscript titled “A feasibility study demonstrating that independence, quality of life, and adaptive behavioral skills can improve in children with Down syndrome after using assistive technology” to PLOS ONE. We are grateful for the additional time and effort that you and the reviewers have dedicated in providing valuable feedback on our initial revised manuscript. Below are the point-by-point responses to the reviewers’ comments and concerns: 

Comments from Reviewer #2

 Comment 1: “Abstract: The term ’nationally’ implies a reader knows where the study is conducted. Consider naming the country instead.”

Response: We agree with the reviewer that ‘nationally’ is vague for an international audience. We have replaced all mentions of the word to the specific country: United States of America. 

Comment 2: “Personally, I find it difficult to use the framing ‘the first study on X’” 

Response: We understand the reviewer’s caution with framing a study as the first to investigate on a particular subject. We have reframed the study as contributing to the expansion of research in an area that is still relatively new. 

Comment 3: “I think the term significant should not be stated in the question”

Response: Following the reviewer’s suggestion, we have removed the word ‘significant’ to reserve its use for discussing statistical outcomes. 

Comment 4: “Introduction: I think the outline lacks a clear description of what is meant by feasibility of AT…Additionally, there is little to no description of what kind of AT that is the focus of the study, and why you choose to study the MapHabit System”

Response: We have included additional details that define feasibility for the scope of our study and provided further rationale for using the MHS for the intervention. 

Comment 5: “I would have preferred describing and provide the rationale for each construct/the feasibility take in the Introduction section”

Response: To clarify the structure of the manuscript, we have made changes in the Introduction. In addition, we included new subheadings to the Method section, as suggested by the reviewer in the next comment. 

Comment 6: “Method section. General: the Enrollment, consent, & clinical trial registration section would be a lot easier to read if there were subheadings.”

Response: We have added subheadings to this section for better organization. 

Comment 7: “L54: Name the country.”

Response: The country (USA) is now named in the manuscript. It is now located in line 78.

Comment 8: “L115…I don’t think the design allow for them being their own controls just because you measure pre- and post-intervention…Consider removing that argument or provide a clearer description, number of measurement points etc.”

Response: We appreciate the reviewer’s comment. The sentences were removed and replaced to be more straightforward: “permitted us to obtain baseline measures and compare them with the post-intervention outcomes”. 

Comment 9: “L130-135: Provide rational of why you used non-parametric tests…Also, in the results section you write: ‘one-sample Wilcoxon test’. Is this correct”

Response: 

We appreciate Reviewer #3’s discussion of the statistical aspects of the study. The Reviewer is correct, the test used for the analysis on line 136 should have been indicated as the Wilcoxon Signed Rank test, not the t-test. We have corrected the typo. 

Additionally, Reviewer #3’s query about why nonparametric tests were used for the analyses (particularly for the QOL analysis) encouraged a second look. For the present study, the data sets were generated from relatively small samples (and possibly skewed data, because of the nature of the diagnosis of the participants). Accordingly, it was likely that our samples would yield scores from dependent measures that would depart from normality and potentially violate assumptions inherent to parametric tests (Seigel, 1956; Leech et al., 2002). Additionally, small sample sizes, as in the present study, will not cause the results derived from nonparametric analyses to be misleading to the extent that small sample sizes and outliers unduly can affect parametric tests (Hollander and Wolfe, 1973; McSeeney and Katz, 1978). Taking these points together, we feel confident our use of nonparametric tests in this study, including the QOL analysis, is appropriate, and we thank the Reviewer for the stimulating query.

Comment 10: “Method and Results. You have not mentioned the attrition or analysed what that meant for your conclusions.”

Response: We agree with the reviewer that the attrition of our participants is a limitation of the study. This may be explained by the study being too burdensome in the COVID-19 pandemic environment. We have added this point as a limitation in the Conclusion. 

Comment 11: “I think a conflict-of-interest statement should be added to the manuscript.”

Response: Per the guidelines of PLOS ONE, a conflict-of-interest statement has already been submitted to PLOS ONE separately. 

Comments from Reviewer #3

Comment 12: This well written research paper focuses on the use of AT from children with DS - It is suggested authors to give more information about the evaluation of their intellectual disability; the results may differ according the severity.

Response: We believe that the reviewer’s point is a very important one, and further reinforces the importance of this feasibility study. Further studies that are more systematic, such as randomized controlled trials, will allow for more conclusive findings and the ability to stratify the participants by intellectual or developmental disability severity. The present study did not have sufficient power or number of participants. We have added this point as a limitation to our conclusions. 

Sincerely,

Samuel S. Han

Clinical Lead

MapHabit, Inc., Atlanta, GA 30308

shan@maphabit.com

Tel: 207-991-1955

References

Hollander, M., & Wolfe, D. A. (1973). Nonparametric Statistical Methods (Wiley Series In Probability And Mathematical Statistics). Wiley.

Leech, N. L., & Onwuegbuzie, A. J. (2002). A Call for Greater Use of Nonparametric Statistics.

McSweeney, M., & Katz, B. M. (1978). Nonparametric statistics: Use and nonuse. Perceptual and Motor Skills, 46(3_suppl), 1023-1032.

Messick, S. (1957). Nonparametric Statistics for the Behavioral Sciences. Sidney Siegel. McGraw-Hill, New York, 1956. 312 pp. $6.50. Science, 126(3267), 266-266.

---

## [Decision Letter · Decision Letter 2]

6 Mar 2023

PONE-D-22-05428R2A feasibility study demonstrating that independence, quality of life, and adaptive behavioral skills can improve in children with Down syndrome after using assistive technologyPLOS ONE

Dear Dr. Han,

Thank you for submitting your manuscript to PLOS ONE. After careful consideration, we feel that it has merit but does not fully meet PLOS ONE’s publication criteria as it currently stands. Therefore, we invite you to submit a revised version of the manuscript that addresses the points raised during the review process.

The manuscript has been evaluated by three reviewers, and their comments are available below.

Although reviewer 3 is now satisfied with the revised manuscript, reviewers 1 and 2 still have some requests for clarification.Reviewer 1 requests some minor language edits with reference to the Wilcoxon test, and reviewer 2 would like additional detail regarding their earlier requests.Could you please revise the manuscript to carefully address the concerns raised?

We look forward to receiving your revised manuscript.

Kind regards,

Steve Zimmerman, PhD

Associate Editor, PLOS ONE

Journal Requirements:

Reviewers' comments:

Reviewer's Responses to Questions

**Comments to the Author**

1. If the authors have adequately addressed your comments raised in a previous round of review and you feel that this manuscript is now acceptable for publication, you may indicate that here to bypass the “Comments to the Author” section, enter your conflict of interest statement in the “Confidential to Editor” section, and submit your "Accept" recommendation.

Reviewer #1: All comments have been addressed

Reviewer #2: (No Response)

Reviewer #3: All comments have been addressed

2. Is the manuscript technically sound, and do the data support the conclusions?

Reviewer #1: Partly

Reviewer #2: Partly

Reviewer #3: Yes

3. Has the statistical analysis been performed appropriately and rigorously? 

Reviewer #1: Yes

Reviewer #2: No

Reviewer #3: (No Response)

4. Have the authors made all data underlying the findings in their manuscript fully available?

Reviewer #1: Yes

Reviewer #2: No

Reviewer #3: Yes

5. Is the manuscript presented in an intelligible fashion and written in standard English?

Reviewer #1: Yes

Reviewer #2: Yes

Reviewer #3: Yes

6. Review Comments to the Author

Reviewer #1: Line 154, Line 177 and Figure 2, the name for one-sample Wilcoxon test or one-sample Wilcoxon signed-rank test to be standardised or alternatively one-sample Wilcoxon test (also referred as one-sample Wilcoxon signed-rank test) to be indicated.

Reviewer #2: I cannot see h0w you have addressed some of the concerns I raised. Responses for comment 4 and 5 say you have added rationale and descriptions in the manuscript, which I cannot find. Also, the attrition is only mentioned as a limitation but no one knows how these participants' scores would have altered the conclusions. Still, the abstract does not mention this uncertainty.

Reviewer #3: (No Response)

7. PLOS authors have the option to publish the peer review history of their article (what does this mean?). If published, this will include your full peer review and any attached files.

Reviewer #1: No

Reviewer #2: No

Reviewer #3: **Yes: **Aspasia Serdari, Assoc Professor of child Psychiatry, Medical School, DUTH

---

## [Author Response · Author response to Decision Letter 2]

22 Mar 2023

Dear Dr. Zimmerman, 

Thank you for giving us the opportunity to submit a revised manuscript titled “A feasibility study demonstrating that independence, quality of life, and adaptive behavioral skills can improve in children with Down syndrome after using assistive technology” to PLOS ONE. We are grateful for the additional time and effort that you and the reviewers have dedicated in providing valuable feedback that will enhance the manuscript. Below are the point-by-point responses to the reviewers’ comments and concerns: 

Comments from Reviewer #1

 Comment 1: Line 154, Line 177 and Figure 2, the name for one-sample Wilcoxon test or one-sample Wilcoxon signed-rank test to be standardised or alternatively one-sample Wilcoxon test (also referred as one-sample Wilcoxon signed-rank test) to be indicated.

Response: We thank the reviewer for noting the inconsistency of naming the statistical analysis. We have clarified this by following the reviewer’s suggestion: “one-sample Wilcoxon test (also referred to as one-sample Wilcoxon signed-rank test)”. 

Comments from Reviewer #2

Comment 1.A: I cannot see how you have addressed some of the concerns I raised. Responses for comment 4 and 5 say you have added rationale and descriptions in the manuscript, which I cannot find. 

Response: We believe comment 4 and 5 from the previous review have been addressed in the last rendition of the manuscript, seen in the Introduction (line 38-40; 48-50; 53-54) and Methods (line 100-105; 109-115). 

Comment 1.B: Also, the attrition is only mentioned as a limitation, but no one knows how these participants' scores would have altered the conclusions. Still, the abstract does not mention this uncertainty.

Response: We appreciate the reviewer noting the potential limitation of the incompletions seen in participation. We have clarified this as a limitation that could have affected the outcomes that we reported. This has also been addressed in the abstract.

---

## [Editor Report · Decision Letter 3]

10 Apr 2023

A feasibility study demonstrating that independence, quality of life, and adaptive behavioral skills can improve in children with Down syndrome after using assistive technology

PONE-D-22-05428R3

Dear Dr. Han,

We’re pleased to inform you that your manuscript has been judged scientifically suitable for publication and will be formally accepted for publication once it meets all outstanding technical requirements.

Kind regards,

Vanessa Carels

Staff Editor

PLOS ONE
---

## [Editor Report · Acceptance letter]

15 May 2023

PONE-D-22-05428R3 

A feasibility study demonstrating that independence, quality of life, and adaptive behavioral skills can improve in children with Down syndrome after using assistive technology 

Dear Dr. Han:

I'm pleased to inform you that your manuscript has been deemed suitable for publication in PLOS ONE. Congratulations! Your manuscript is now with our production department. 

Kind regards, 

on behalf of

Dr. Vanessa Carels 

Staff Editor

PLOS ONE